# Costing the intergovernmental negotiating body (INB) process

**Clare Wenham** [1] *, **Akhila Potluru** [2]

**1** Department of Health Policy, London School of Economics and Political Science (LSE), London, United Kingdom, **2** LSE Health, London School of Economics and Political Science (LSE), London, United Kingdom

* c.wenham@lse.ac.uk

**Data Availability Statement:** All data are available and included in supplementary file.

**Funding:** The authors received no specific funding for this work.

## Abstract

The Intergovernmental Negotiating Body (INB) was established to draft and negotiate a pandemic instrument to strengthen pandemic preparedness and response (PPR). This has been carried out under the auspices of the World Health Organization (WHO), and has to date involved 15 sessions in Geneva, plus countless hours both in informal working groups, and in capitals working on government positions on each substantive issue. This all comes at a cost, both in terms of human resource and travel costs associated with the development of an international treaty and its associated process. We begin to quantify the cost of this process as approximately US$ 201,343,032. If we also consider estimated costs for the parallel WGIHR process to be US$56,024,830, we estimate the total cost of this combined governance development to be US$257,367,862. We position this in the context of how much pandemic preparedness is thought to cost on an annual basis, and the opportunity costs of investing in this governance process instead of more operational areas of health security. Moreover, in doing so, we offer one of the first financial estimates of the cost of developing and negotiating multilateral treaties.

## Introduction

The Intergovernmental Negotiating Body (INB) was established by a Special Session of the World Health Assembly in November 2021 (WHASS), with the mandate to draft and negotiate a convention, agreement or other international instrument to strengthen pandemic prevention, preparedness and response by May 2024 [1]. Since then, governments have met numerous times, both formally in plenary sessions hosted at WHO in Geneva, and informally online and in person to determine the contents of this agreement, and overcome the differing political and technical positions of governments on issues as wide ranging as One Health, pandemic financing, and access and benefit sharing. Despite considerable time and effort in May 2024, there remained significant gulfs between differing government positions and there was little time to negotiate meaningfully and reach a consensus text. Instead, at the World Health Assembly, member states agreed to extend the mandate of the INB by a further year until May 2025. At the same time, Member States of WHO have also agreed, through the efforts of the 2 year long Working Group for International Health Regulations (WGIHR) to targeted

**Competing interests:** I have read the journal's policy and the authors of this manuscript have the following competing interests: CW reports consulting fees from WHO EURO, but not as part of the INB process. CW also served on the IHR Review Committee for amendments to IHR.

amendments of the International Health Regulations (IHR) (2005) the legally binding regulatory framework which exists to prevent, protect against, control and provide a public health response to the international spread of disease in ways that are commensurate with and restricted to public health risks, and which avoid unnecessary interference with international traffic and trade.

This all comes at a cost. This includes the costs of hosting meetings in Geneva, the human resource cost of having teams of civil servants working on positions for each issue area, not to mention the opportunity costs of focusing efforts on negotiating a new legal instrument, rather than operationalising other areas of PPR, or indeed advancing other policy areas within health. We sought to quantify these costs of negotiating the pandemic agreement.

We were unable to find a methodology to suitably cost the process of starting international negotiation from scratch. The Summary Records of the World Health Assembly's Special Session in November 2021, which established the INB, estimated that the total resource requirements for the Secretariat to implement the process would be US$2.84million. This was a proposed budget, and it only considered the costs within WHO itself, and not for Member States. This number is problematic, as it budgeted no staff time for WHO staff, and offers no detail as to how the number was reached [2]. At WHA77 in May 2024, member states agreed to extend the mandate of the INB to continue to negotiate the pandemic agreement. At this time, the WHO suggested the financial implications for the Secretariat would be US$5.12million, albeit whilst noting this was only 4 months' worth of costs [3].

Existing literature estimating the cost of conferences focuses on private sector events, which fail to account for the security arrangements, diplomatic protocols and the wide array of government involvement in inter-governmental meetings [4]. Some estimates of costs of inter-governmental meetings exist, such as BWC which estimate that their annual meetings cost US$610,000 per annum, albeit they do not offer a methodology for calculations [5]; WHO estimate that a 4 day meeting for governance reform costs approximately US$458,826 [6]; UN estimate that the average cost of a UN event is US$1.7–3.4million per event, and up to US$10M for the Rio Earth Summit. However, these costings are exclusively for one meeting, rather than multiple meetings over the course of several years, and with person time within governments in fleshing out substantive content for provisions in between. As such, we believe that this is among the first studies to provide a detailed financial estimate of treaty-making in global health. This matters as it allows us to open the conversation about whether this is a good use of money, particularly within a context of scarce resources and dwindling political commitment to increase financing for pandemic preparedness and response.

## Methodology

Our methods focused on three key areas: human hours put towards the treaty process (in terms of a percentage of full-time salary of civil servants); costs of states sending delegates to in person meetings in Geneva; the cost of hosting INB sessions in Geneva.

### Human hours

The calculation of human costs involved a systematic approach to calculating the remuneration for delegates based on their roles, categorisation, and estimated time allocation towards the treaty proceedings. Given the behind closed doors approach of the INB, those involved in the process were identified from their participation in the Special Session of the World Health Assembly [7]. Delegates were categorised by their given roles of Chief Delegate, Deputy Chief Delegate, Delegate, Alternative and Advisor. This provided the number of delegates per country, how many people were in each given role, and also whether they were a capital official

(non-Geneva delegate) or a permanent mission official (Geneva delegate). Salary data for government officials was sourced from the World Salaries database (accurate as of June 2024) [8]. Chief Delegates were assigned the highest civil servant salary for their respective countries. Deputy Chief Delegates were assigned a salary midway between the highest and average civil servant salaries. Delegates and Alternates were allocated the average civil servant salary. For Geneva-based delegates, salary estimates were derived from the United Nations Common System of Salaries, Allowances, and Benefits [9]. Salaries were categorised based on professional grades, where chief ambassadors were D2 level, and deputy chief ambassadors were either P5 or D1 as advised by UN guidance. Further adjustments were made according to the country's average civil servant salary compared to the global average.

Given the multifaceted responsibilities of delegates beyond the pandemic treaty process, an adjustment was made to reflect the proportion of their working hours dedicated specifically to treaty-related activities. Non-Geneva Chief and Deputy Chief Delegates were assumed to spend 10% of their time on treaty proceedings. Non-Geneva Delegates, Alternates, and Advisors were assumed to allocate 25% of their time to the treaty process. All Geneva delegates were assumed to allocate 50% of their time to the treaty process. We verified such assumptions with four-member state delegates participating in the process (see limitation section below).

We calculated the costs of WHO Secretariat participation in the process. Again, noting the lack of transparency of the INB, we estimated this to be the equivalent to 5 staff working full time [1 each at P1, P2, P3, P4, P5] [10] for the full duration of the INB process at HQ (Geneva) and one each at regional level.

Finally, we considered the costs of representatives of CSOs participating in the INB process. To do so, we included all NGOs which are in official relations with WHO and whom participated in the WHASS [7], and assumed that each representative is paid the average non-profit campaign manager salary for the country that the CSO is headquartered in [8].

## Sending delegates to in-person INB sessions in Geneva

The estimation of costs incurred by countries to send delegates to INB meetings in Geneva involves several components: travel, accommodation, food, and other expenses. To estimate the average cost of return flights from each country to Geneva, data from booking.com was provided a reasonable approximation of the expenses incurred. The average flight cost was converted into United States Dollars (USD) for uniformity. To assess per diem costs (or otherwise) the United Nations Development Programme (UNDP) Daily Subsistence Allowance (DSA) rates for Switzerland were adopted. These rates encompass lodging, meals, transport, and other relevant expenses. DSA rates from 2022 to 2024 were collected to ensure the correct rate was applied to each year. The total cost for each country to send delegates was calculated by multiplying the length of the trip (INB meeting length plus 2 nights) by the DSA rate of the respective year and adding the average return flight cost to Geneva (one flight per INB/resumed session). We costed WHO participation from those in regional offices, but not those based at HQ, noting that they were Geneva based anyway. For CSO representatives, we assumed that only 20% of representatives attended all INB sessions (and note this as a placeholder for the variation between eg. European Union and smaller disease specific NGO actors). For all figures, Swiss staff are excluded as they do not require foreign travel or a separate DSA allowance to attend INB sessions.

## Calculating cost for WHO to hold INB sessions in Geneva

The estimated minimum costs of the "Meetings on Governance Reform"[6], served as a reference point for estimating the costs of hosting the INB sessions. The cumulative inflation rate

for Switzerland was applied to 2015 costs to provide a more accurate assessment for the INB sessions. We conducted a literature review to identify potential additional expenses associated with the pandemic treaty process. These additional cost factors were incorporated into a comprehensive 'master list' of costs for further analysis.

Costs were categorised into fixed costs, which are incurred once per session, and variable costs, which increase with the duration of the INB sessions to facilitate a more accurate calculation of the overall event expenses. The total cost for each INB session was computed by summing the fixed costs with the product of variable costs and the length of the session. Subsequently, the total cost for all INB sessions was determined by aggregating the costs of individual sessions.

### WGIHR

We sought to consider these costs of the INB along that of the WGIHR, the parallel process which is ongoing in Geneva to enhance pandemic preparedness and response governance. We calculated the costs for the parallel process ongoing of the amendments to the International Health Regulations (2005), using the same methodology, but with less human resource in capital dedicated to it, specifically all delegates (both Geneva and Non-Geneva) are assumed to only spend 25% of their time, noting these were targeted amendments to an already existing regulatory framework, rather than the creation of a new treaty text.

### Sensitivity

We performed a sensitivity analysis on our overall figures to offer the potential for error in the assumptions used for our calculations. We present each of the combined costs with a 5% 10% and 25% margin of error each way, to determine how variability in human hours, salary, event/travel costs may impact our final figures.

## Results

### Human cost

The adjusted salaries for both Geneva and non-Geneva delegates were summed to compute the total human cost per country over the three-year duration of the pandemic treaty process, for INB sessions 1 to 15. This accounted for a total of US$124,229,337. The human costs of WHO staff we estimated to be US$5,101,224. The total human hours costs for CSO representatives over 3 years was US$34,506,790. A total human cost of the INB, therefore, **equals US $163,837,351.**

### Sending delegates

The total estimated costs of transport and daily standard allowances for delegates, assuming that DSA is given to all delegates, and assuming two delegates per meeting totals US $27,261,646. The cost of sending full-time WHO staff to all INB sessions amounts to US $568,892. For CSO representatives, the cost is US$4,333,764. Combined, this gives us a total cost of sending delegates, WHO staff, and CSO representatives to in-person INB sessions of **US$32,164,302**

### Meeting cost

The total estimated cost of holding all meetings in Geneva at WHO is US**$5,341,379.**

## Total INB cost

This brings the estimated total cost of the INB process so far to be US$201,343,032—or equivalent to US$13,422,869 per meeting.

## WGIHR cost

We estimated the WGIHR hosting cost WHO US$1,761,080; the estimated human hours to be US$42,684,966; and travel and associated costs to be US$11,578,784 (noting that often these meetings were held back-to-back with INB meetings which may reduced some of these costs, not reflected here). The total cost for WGIHR is estimated to be US**$56,024,830.**

## Total for WGIHR and INB

The total for INB and WGIHR we estimate to **be US$257,367,862.**

Given that our final estimate is based on a range of assumptions, we provide sensitivity analysis of a 5% 10% and 25% margin of error each way to account for variability in each contributing factor, following best practice [11]. When taking each margin of error in turn, our final estimate of US$201,343,032 for INB sessions 1 to 15 is altered as follows:

±5% equates to ± US$10,067,151 leading to upper and lower estimates of US$211,410,184 and US$191,275,881 respectively.

±10% equates to ± US$20,134,303 leading to upper and lower estimates of US$221,477,336 and US$181,208,729 respectively.

±25% equates to ±US$50,335,758 leading to upper and lower estimates of US $251,678,791and US$151,007,274 respectively.

## Limitations

We believe our figures to be almost certainly under-estimating the true cost of the INB (and WGIHR) process(es) to date. Firstly, we have made considerable assumptions on proportion of time devoted to the INB and in turn to the WGIHR, and this may not capture real world variations, and time spent over the three-year process. We anticipate percentage time as an average per year, noting that this may fluctuate depending on workflow, timing of meetings and member state's particular interest in health security. Moreover, standardised salaries and variations in remuneration policies for each delegate/country could introduce discrepancies in salary estimations delegates. Sensitivity analysis has helped to contextualise these figures and the range of potential costs.

Secondly, we have made assumptions on who carries out the work, using the delegate list of the World Health Assembly Special Session, which may not reflect the true burden of activity within Geneva or indeed within capitals. Moreover, we have also only included those who might be directly tasked with INB efforts. This would not account for those elsewhere in ministries who work on topics which are affiliated to the contents of the treaty, and who may contribute to specific provisions or contents discussions and recommendations. Accurately detailing this would be impossible, given the scope of the treaty and its contents, and indeed the variability of capacities of national (and sub-national) departments. We were also unable to estimate the hours that CSO and WHO teams dedicated to the INB process and have made assumptions about how many of these organisations have participated throughout the process. Secondly, we have assumed the physical presence of one or two delegates per country at each INB meeting, however actual attendance almost certainly varied, with this being conducted online, or solely by Geneva based staff. Using average flight costs may not accurately reflect the specific expenses incurred by each country, considering fluctuations in airline fares and travel

seasons. We have used the lower estimates by assuming they will fly economy, but actual costs may be higher as some delegates may be flying business class. Again, sensitivity analysis has allowed these numbers to be placed in a spectrum of potential costings. While UNDP DSA rates were utilised to standardise meal and hotel costs, variations in country allowances, for travelling delegates may result in discrepancies between estimated and actual expenses.

Third, the costing documents from WHO in 2021 and 2024 give some indication as to what costs the secretariat believe to have borne, but we believe these may be an underestimate also. It is possible that certain expenses within WHO were overlooked. Finally, although we applied inflation adjustments to the 2015 costs, variations in cost structures and economic factors over time may not be fully captured by this approach.

## Discussion

US$201 million is a considerable sum of money in a context of constrained resources for health and in particular pandemic preparedness in the wake of COVID-19 and other recent epidemics. The relative costs of the process are also not equitably distributed amongst member states, and this in turn impacts engagement from different states in the process itself, practically and substantively. We consider the cost of this in the context of financing pandemic preparedness and response, and as opportunity costs and in the broader pandemic governance landscape.

Several groups have previously sought to estimate the costs of achieving meaningful pandemic preparedness [12–15]. These vary drastically, based on what is being counted within pandemic preparedness and response, with figures ranging from US$10bn to US$204bn globally per annum, and an average of US$31.8bn per year. In this context, the cost of the INB process represents 0.63% of the estimated costs of pandemic preparedness. The pandemic treaty is a key part of a rule-based system international system, for which governments have worked hard to maintain order and coherence, and from that perspective, such an investment can be seen as part of a necessary expenditure to maintain governance and developments in international law. Indeed, the Independent Panel for Pandemic Preparedness and Response highlighted that investments for pandemic preparedness and response would require adjustments to governance structures [16], and this is such a process deemed important by many in the global health architecture.

The financial costs of the INB should also be contextualised against the potential for policy impact when comped to the limited gains of treaty making in general, and in particular the other multilateral treaty making initiatives under WHO [17]. WHO has only used its extensive treaty making powers twice before, for the negotiation of the International Health Regulations in 1969 (and latterly their revisions in 2005); and for the creation of the Framework Convention for Tobacco Control (FCTC) in 2003. Whilst the IHR (2005) for the most part is deemed to be a success; creating a shared normative understanding of how best to mitigate the transnational spread of disease, doing so whilst also preventing unnecessary impact on travel and trade, they are also much more limited in scope than the proposed pandemic agreement. Moreover, there have also been multiple tensions within their operationalisation, not least, its limited implementation, with states in 2023 averaging 64% compliance with their obligations [18]; a lack of financing available to support implementation; and the multiple issues which arose with IHR effectiveness during previous health emergencies [19]. Meanwhile, the FCTC has arguably strengthened tobacco control, for example with enhanced labelling, and bans on marketing tobacco products. However, it has not reduced global cigarette consumption in the years since its adoption, with regional differences in decreased consumption in HIC and increased consumption in LMICs [20]. It has also been hampered by major tobacco producing

states not ratifying the convention and private sector interference, in turn limiting its effectiveness. As such, there is a broader question as to the normative and legal value of negotiating a pandemic agreement, regardless of the costs associated with this activity. The alternative argument is that the process itself of negotiating a treaty produces a normative understanding of expectations of behaviour between states, and fosters trust between states and in the multilateral system, despite whether these processes result in successful treaties per se.

We could alternatively consider the cost in comparison to the estimates for strengthening global health institutions including the WHO, the host institution of the INB. This has been a frequently touted activity globally, and something which is seen as vital to pandemic preparedness. Such costs have been estimated to be US$5bn per year, for which the INB represents 4%. However, the money spent on the INB has not been used for directly strengthening WHO, or fostering investment in resources for the organisation, or for even financing the hosting the INB Secretariat. Indeed, it could be argued that the hosting of the INB has in fact done more to weaken the WHO institutionally and politically, rather than strengthen it. There was much debate when the INB was established as to whether it should be hosted in Geneva under WHO or in New York under the United Nations. The WHO pushed for it to be the forum for negotiations, but the relative inexperience of the institution in negotiating international agreements, as well as the fact that it is still suffering from a legitimacy crisis in the wake of the handling of COVID-19 has been problematic. This perceived weakness may have been the reason that member states selected the WHO for the negotiations, noting that they would not be tied to any future obligations within a weaker governance structure [21]. The fact that pandemic treaty was not agreed by the deadline of May 2024 under the auspices of WHO, combined with anecdotal rumours of an overly involved WHO secretariat in what was supposed to be a member state led process may do more reputational and institutional damage than had the negotiations not taken place in Geneva at all. Thus, the spend on the INB has arguably not contributed to a strengthened WHO, despite member states' continual rhetoric of wanting an empowered institution. Lessons here could be learned from other multilateral for a who are more used to managing such negotiations on a regular basis, such as other UN entities. For example, processes and best practice for such member state led, institutionally supported negotiations have been well established over centuries, and are commonplace in diplomatic circles [21–23]. These could be better embedded in WHO approaching the pandemic treaty from the start to ensure the most successful outcomes.

A second area for concern with WHO is the lack of financing available for the effort that they have undertaken. As WHA77/A77 noted, there is a financing gap for the work of the INB secretariat, and that WHO was continuing to look for pledges and funding for the activities associated with the work of negotiating a pandemic agreement. This has to be considered within the context of a continually underfunded WHO, with many member states failing to capitalise their GDP commitments into hard cash for the organisation's assessed contributions which could finance this administrative burden, and so the WHO is having to seek voluntary contributions to perform the functions asked of it by the very same member states that fail to pay their own dues. The fact that there is not funding in place for this continued activity is of considerable alarm, noting the potential impact this could have on WHO performing the function of the host institution and secretariat, and the risk that performance limitations will have on the institution reputationally. Anecdotally we understand that some INB sessions have had to be curtailed because WHO was unable to continue to pay the translators overtime, it highlights the distinct budgetary challenges that exist within the very institution amid which the INB process is occurring.

Indeed, we could compare these costs in relation to member state contributions to WHO. Each member state of WHO is required to contribute a percentage of the country's GDP as

agreed by the UN General Assembly, and these come in the form of assessed contributions to WHO. In turn, member states can provide further financing to the institution by way of voluntary funds, to contribute to programmes which align with their donor interests, or as contingency funds in the case of an emergency. In this instance, the INB process equates to approximately 3.0% of the Biennium Programme Budget 2022–3 (US$6.72bn), and approximately 2.9% of the Biennium Programme Budget 2024–5 (US$6.83bn) [24]. As an exemplar, the total estimated INB spend is far greater than the total funding (assessed, voluntary and contingency funds) given to WHO in 2024–5 by the United Kingdom (US$177,395,000) [25].

We note that even in the case of ±25% error margin, the upper bound only greatly exacerbates the severity of the points made below in the discussion, and the lower bound estimate of US$151,007,274 is still an incredibly high figure, bearing in mind that this is broadly similar to Japan's total contributions to WHO of US$147 million [26]. Therefore, even in the unlikely case where our total estimate is subject to a 25% error margin, it still merits debate and evaluation on whether this is an efficient allocation of increasingly limited resources for global pandemic preparedness.

We recognise that the large share of this money has not come from WHO per se, but from Member States, this is here simply as an illustrative point, in demonstrating the relative costs of the INB compared to the broader WHO funding landscape, to demonstrate the considerable proportion of spend that it equates to when compared to the broad remit of work occurring across the organisation.

However, assessing the opportunity costs of the pandemic treaty forces us to ask whether pandemic preparedness and response could be strengthened more by allocating the money differently. For example, whether conceptual preparatory governance measures are cost effective at the expense of operational implementation of preparedness and response mechanisms, such as direct contributions to health system strengthening or support for skilled health and public health workers. For example, the cost of the INB process to date could equate to 120.5 million COVID-19 vaccine doses in arms. Similarly, this cost could equate per year to 871 doctors and 2,493 nurses in USA; 883 doctors or 2,740 nurses in UK; 17,586 doctors or 56,229 nurses in India; 8,508 doctors or 29,180 nurses in Philippines; 3,805 doctors or 12,401 nurses in Poland; 3,726 doctors or 12,005 nurses in South Africa; 11,272 doctors or 34,855 doctors in Zimbabwe; 3,801 doctors or 12,229 nurses in Brazil [8]. This raises concerns regarding the efficiency of its resource utilisation, and what best spends could be in pandemic preparedness and response. We believe our analysis which starts to quantify the overall costs of this process can lead to broader conversations about whether there needs to be more detailed budgets in place before multilateral negotiations are agreed to. The only public data that we have is the significantly underestimated proposal by WHO for their internal secretariat costs presented at WHA/SS in November 2021. It is possible that member states undertook their own domestic budget assessments prior to this session which in turn influenced their position as to whether to proceed or not, but we are not privy to such information. Moreover, these would likely only relate to the cost of participation for their diplomats or civil servants, and there would not have been a sense of the overall cost of the whole process to make a value judgement as to whether it would be a good use of limited resources in pandemic preparedness and response, particularly given the mixed evidence of effectiveness of other multilateral treaties in global health.

Detailing such costs of a multilateral process may have repercussions for future pandemic preparedness governance. For example, the current proposals of the pandemic agreement include the proposed creation of a Conference of Parties for the development and implementation of the pandemic agreement, the costs associated with which will be like those we have detailed above. Similarly, there is growing discussion that the pandemic agreement will be complemented with a series of further protocols which will outline greater operationalisation

of pathogen access and benefit sharing (PABS), financing and One Health. Negotiations for each of these contentious items will likely accrue parable costs as we have detailed above.

Moreover, the diversion of resources and political momentum towards the treaty negotiations, coupled with the lack of tangible outcomes, could lead to scepticism regarding the efficacy of investing in similar multilateral initiatives. Failure to achieve consensus on a comprehensive, meaningful pandemic agreement may also undermine international cooperation and solidarity in this forum, and governments willingness to continue with associated protocols or Conferences of the Parties. This is particularly important for low-income countries. Whilst some intergovernmental process, such as the WHO meetings on governance reform have offered assistance in travel expenses for low-income countries, the INB has not published information stating that it has provided such support. Indeed, the only information provided in official documents notes that WHO would cover the costs of members of the Bureau of the INB, and WHA76/34 notes the establishment of a Voluntary Health Trust Fund for small island developing states to support the participation of their representatives in formal meetings and negotiations [27,28]. However, this is only due to be launched in 2024/5, as such we do not believe it will have contributed to financing for the INB thus far. Thus, we assume that the costs of the INB are particularly acute for those in LMICs, and/or they have not been able to participate to the extent that high income countries have, noting also that the convention has taken place in Geneva, to which travel is relatively cheaper for example, if coming from Europe, compared to the African continent, thus further exacerbating financial strain on low-income countries. Moreover, it is smaller delegations that may not have permanent representation in Geneva, further limiting their participation in the process. This disparity is particularly gruelling as the outcomes of the INB have yet to demonstrate tangible benefits for low-income countries, as evidenced by issues such as access and benefits sharing, financial support for pandemic preparedness and response and operationalisation of equity. Whilst all INB sessions have also been hosted in a hybrid function, this may not be a pertinent way forward for meaningful diplomatic negotiations, and this penalises those not in the room. Given that WHO has 6 regional offices, costs could be dissipated more equally through the rotation of the hosting of INB meetings in each location, which would in turn reduce costs in part, and allow for equitable distribution of working hours and travel time/time away from home.

In terms of broader costs, many governments have further pointed out the challenges of the process for work/life balance, including the mental health of delegates who have had to work across working hours, missing family commitments and other work obligations. This is in the context of a tight timeline, and large plenary meetings where given the nature of consensus building, progress has been slow, and it has been challenging for smaller delegations where parallel work streams have developed. We do not yet know what the impact of this might be on workforce retention, and productive outputs within government services.

This figure does not account for informal or separate intergovernmental meetings that discuss the substantive content of pandemic treaty. Furthermore, it is likely that much of the substantive content of the current drafts of the pandemic agreement text, such as those of Pathogen Access and Benefit Sharing (PABS), financing and One Health might move into their own protocols or instruments, as well as a sense that many governments now the IHR have completed are pushing for a framework convention and/or for the content of the pandemic treaty to be non-binding. Such outcomes weaken the overall rule-based system of governance, and weaken future obligations on states to enhanced pandemic preparedness and response. As such, amid the political questions which remain about the process from here on in, the substantive content and negotiation positions, further questions should be asked about its value for money. We begin to quantify such costs for the readers, negotiators, and beyond, in an effort to ensure that global taxpayers money is being spent in the most beneficial way for

the future of global health security. Moreover, we hope such costing estimates may guide future governance and multilateral negotiation processes in global health, to ensure that limited resources and allocated efficiently, striking a balance between governance needs and operational activities. Whilst ultimately the normative dimension is required to set the strategy and standards for the implementation of the operational, these can be a zero-sum game in a time of neglect in pandemic preparedness and response efforts [29,30]. Whilst there remain multiple operational demands from previous governance approaches and failures during COVID-19, this balance between normative and operational may need to be re-considered.

## Author Contributions

**Conceptualization:** Clare Wenham.

**Data curation:** Akhila Potluru.

**Formal analysis:** Clare Wenham.

**Funding acquisition:** Clare Wenham.

**Investigation:** Clare Wenham.

**Methodology:** Clare Wenham, Akhila Potluru.

**Project administration:** Clare Wenham.

**Resources:** Clare Wenham.

**Supervision:** Clare Wenham.

**Validation:** Clare Wenham.

**Writing – original draft:** Clare Wenham.

**Writing – review & editing:** Clare Wenham, Akhila Potluru.

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
