## [Decision Letter · Decision Letter 0]

25 Sep 2024

PGPH-D-24-01747

Costing the INB Process

Dear Dr. Wenham,

Thank you for submitting your manuscript to PLOS Global Public Health. After careful consideration, we feel that it has merit but does not fully meet PLOS Global Public Health’s publication criteria as it currently stands. Therefore, we invite you to submit a revised version of the manuscript that addresses the points raised during the review process.

We are pleased to inform you that the paper has been well-received, with one reviewer recommending acceptance, highlighting the importance and timeliness of your analysis of the financial costs of the Intergovernmental Negotiating Body process.

However, the second reviewer has identified several areas where the manuscript could be strengthened to enhance its clarity and robustness. While they appreciated your valuable contribution to global health governance, they have suggested a number of substantial revisions to refine the methodology and expand on the policy implications of your work. 

We believe that addressing these points will significantly strengthen your manuscript and enhance its contribution. We encourage you to carefully consider these suggestions in your revisions.

We look forward to receiving your revised manuscript.

Kind regards,

Elize Massard da Fonseca, Ph.D.

Academic Editor

Journal Requirements:

Additional Editor Comments (if provided):

Reviewers' comments:

Reviewer's Responses to Questions

**Comments to the Author**

1. Does this manuscript meet PLOS Global Public Health’s publication criteria? Is the manuscript technically sound, and do the data support the conclusions? The manuscript must describe methodologically and ethically rigorous research with conclusions that are appropriately drawn based on the data presented.

Reviewer #1: Yes

Reviewer #2: Partly

2. Has the statistical analysis been performed appropriately and rigorously?

Reviewer #1: Yes

Reviewer #2: Yes

3. Have the authors made all data underlying the findings in their manuscript fully available (please refer to the Data Availability Statement at the start of the manuscript PDF file)?

Reviewer #1: Yes

Reviewer #2: Yes

4. Is the manuscript presented in an intelligible fashion and written in standard English?

Reviewer #1: Yes

Reviewer #2: Yes

5. Review Comments to the Author

Reviewer #1: Thank you for the opportunity to review this manuscript - which I genuinely enjoyed reading.

What a fabulous paper and wonderfully creative research area, which offers a different perspective on INB proceedings. I greatly enjoyed reading this paper. There is enough logic in the calculations to support its publication, as one of sufficient methodological-rigor.

Accept.

Reviewer #2: Peer Review Report for Costing the INB Process

Thank you for the opportunity to review this manuscript. I’ve taken the time to carefully go through the paper and supporting documents, and I’m providing feedback to help strengthen the analysis and maximize its impact.

Overall, I like this paper. It’s timely and relevant, offering an important analysis of the financial costs tied to the Intergovernmental Negotiating Body (INB) process. Quantifying these costs is an important step toward understanding how resources are allocated within global governance—particularly in the context of global health. The analysis is especially valuable given the growing conversations about the purpose and utility of international legal agreements, pandemic preparedness, governance structures, and the WHO’s role in all this. The paper stands out by drawing attention to the opportunity costs of these negotiations compared to more direct investments in health security.

However, there are several areas where further clarification and development would enhance the robustness and clarity of the work. I offer the following suggestions:

1. While the paper clearly sets out to quantify the costs of negotiating the pandemic agreement, the introduction could do more to position this research within the literature. The authors mention the lack of prior methodologies, but it would help to explicitly state that this is among the first studies to provide a detailed financial estimate of treaty-making in global health. Also, consider a statement on why this matters—perhaps asking whether we’re truly getting value for money. This would elevate the paper’s relevance and make its contribution to the field more apparent.

2. The approach to estimating human costs is logical and appropriate, using salary data and time allocations. However, it relies heavily on assumptions, as the authors note. This is necessary based on what information is available. But a sensitivity analysis would help demonstrate the robustness of these estimates. For instance, how might the total cost fluctuate if time allocation assumptions (10%, 25%, 50%) change? Additionally, while I understand there are privacy concerns, providing more transparency about the four-member state delegates consulted for these assumptions would strengthen the credibility of the data.

3. The method of calculating travel costs, using average flight costs and per diem rates, makes sense and is well thought out. I appreciate the specificity of return flight costs per country. That said, given the volatility of flight prices, this is another area where sensitivity analysis would be beneficial. A robustness check on these assumptions could provide more confidence in the estimates.

4. A deeper comparison of the INB process with other international negotiation processes would be valuable. How do the costs of the INB stack up against other similar treaties, not only in financial terms but also in terms of their (promised/achieved) policy impact? This could provide important context and strengthen the overall argument.

5. The discussion around opportunity costs is one of the strongest parts of the paper but could be developed more. Could this analysis lead to policy recommendations, such as cost thresholds for international negotiations? Could we imagine more cost-efficient models for treaty-making in global health? Or, is treaty-making itself the most effective (or even a worthwhile) tool for governing global health? These questions would deepen the discussion and make the analysis more forward-looking.

6. The section on the potential negative impact of the INB process on the WHO and challenges faced by low-income countries is compelling but could be expanded. What recommendations could the authors provide to make future negotiations more equitable and cost-effective? Are there strategies to reduce costs while still achieving meaningful outcomes?

7. Rather than focusing solely on the financial costs, the conclusion could end with a broader reflection on how these findings can guide future governance processes. How can we ensure resources are efficiently allocated in global health, striking the right balance between governance and operational needs? Ending on this note may help translate the findings into actionable insights.

Recommendation: This paper makes a significant contribution to global health economics by shedding light on the costs of a major international governance process. However, there’s room to strengthen the methodology, clarify assumptions, and expand the policy implications. I recommend major revisions to address these areas.

6. PLOS authors have the option to publish the peer review history of their article (what does this mean?). If published, this will include your full peer review and any attached files.

**Do you want your identity to be public for this peer review?** For information about this choice, including consent withdrawal, please see our Privacy Policy.

Reviewer #1: **Yes: **Jay Patel

Reviewer #2: No

---

## [Decision Letter · Decision Letter 1]

5 Nov 2024

Costing the INB Process

PGPH-D-24-01747R1

Dear Dr. Wenham,

We are pleased to inform you that your manuscript 'Costing the INB Process' has been provisionally accepted for publication in PLOS Global Public Health.

Best regards,

Elize Massard da Fonseca, Ph.D.

Academic Editor

Reviewer Comments (if any, and for reference):

Reviewer's Responses to Questions

**Comments to the Author**

1. If the authors have adequately addressed your comments raised in a previous round of review and you feel that this manuscript is now acceptable for publication, you may indicate that here to bypass the “Comments to the Author” section, enter your conflict of interest statement in the “Confidential to Editor” section, and submit your "Accept" recommendation.

Reviewer #2: All comments have been addressed

2. Does this manuscript meet PLOS Global Public Health’s publication criteria? Is the manuscript technically sound, and do the data support the conclusions? The manuscript must describe methodologically and ethically rigorous research with conclusions that are appropriately drawn based on the data presented.

Reviewer #2: Yes

3. Has the statistical analysis been performed appropriately and rigorously?

Reviewer #2: Yes

4. Have the authors made all data underlying the findings in their manuscript fully available (please refer to the Data Availability Statement at the start of the manuscript PDF file)?

Reviewer #2: Yes

5. Is the manuscript presented in an intelligible fashion and written in standard English?

Reviewer #2: Yes

6. Review Comments to the Author

Reviewer #2: Peer review report for PGPH-D-24-01747R1

Thank you for the opportunity to review this revised manuscript. I have read the new paper and response to peer review documents.

The authors have done an excellent job addressing and responding to my initial feedback. I am satisfied with their responses. The manuscript should be accepted for publication, pending one very minor comment:

On page 7: I think we are missing a few words: Total INB cost

This brings the estimated total cost of the INB process so far to be $201,343,032 - or equivalent to $13,422,869

I think this should be: 13422869 per meeting/session?

Congratulations to the authors. I think this piece will be welcomed by the community of global health scholars and practitioners, and I see this piece stirring up important conversations.

7. PLOS authors have the option to publish the peer review history of their article (what does this mean?). If published, this will include your full peer review and any attached files.

**Do you want your identity to be public for this peer review?** For information about this choice, including consent withdrawal, please see our Privacy Policy.

Reviewer #2: No
